# Vat-Mediated Mucus Penetration Enables Genotoxic Activity of *pks+ Escherichia coli*

**DOI:** 10.3390/ijms26115353

**Published:** 2025-06-03

**Authors:** Héloïse Chat, Léa Girondier, Guillaume Dalmasso, Caroline Vachias, Laurent Guillouard, Virginie Bonnin, Devon Kavanaugh, Aurélien Birer, Mathilde Bonnet, Nicolas Barnich, Richard Bonnet, Julien Delmas

**Affiliations:** 1Microbes, Intestin, Inflammation et Susceptibilité de l’Hôte (M2iSH), Inserm U1071, INRAE USC 1382, University Clermont Auvergne, 63001 Clermont-Ferrand, France; heloisechat@gmail.com (H.C.); l.girondier@gmail.com (L.G.); guillaume.dalmasso@uca.fr (G.D.); laurent.guillouard@uca.fr (L.G.); virginie.bonnin@uca.fr (V.B.); devon.kavanaugh@uca.fr (D.K.); aurelien.birer@uca.fr (A.B.); mathilde.bonnet@uca.fr (M.B.); nicolas.barnich@uca.fr (N.B.); richard.bonnet@uca.fr (R.B.); 2Institut Universitaire de Technologie Génie Biologie, University Clermont Auvergne, 63172 Aubière, France; 3Institut Génétique Reproduction & Développement (iGReD), University Clermont Auvergne, UMR CNRS 6293-Inserm U1103, 63001 Clermont-Ferrand, France; caroline.vachias@uca.fr; 4Department of Bacteriology, University Hospital of Clermont-Ferrand, 63000 Clermont-Ferrand, France

**Keywords:** colorectal cancer, *Escherichia coli*, colibactin, mucin, *pks*

## Abstract

Colibactin toxin-producing *Escherichia coli* (*pks+ E. coli*) strains are associated with the occurrence of colorectal cancer in humans. These strains induce DNA damage when in close contact with the cells of the intestinal epithelium. Therefore, maintaining the integrity of the mucus layer that covers the intestinal epithelial mucosa is crucial for counteracting the effects of colibactin. The Vat protein is a mucin protease capable of degrading MUC2 mucus proteins that was previously described in adherent and invasive *Escherichia coli* strains. Our work shows that the *vat* gene is found in the genome of all *pks+ E. coli* strains isolated from patients with colon cancer. In mucus-producing HT29-16E cells, we demonstrated that the Vat protein of *E. coli pks+* allows bacteria to penetrate mucus and to reach the epithelial cells. Cells infected with the *E. coli pks + vat-* strain show a reduction in γ-H2AX staining, a marker of DNA damage. Infection of *Apc^Min^*^/*+*^ mice with the *E. coli pks + vat+* strain or the *E. coli pks + vat-* mutant revealed that Vat enhances the ability of *pks+ E. coli* strains to colonize the intestinal mucosa and, in turn, their pro-carcinogenic effects. This study reveals that Vat promotes crossing of the intestinal mucus layer, gut colonization, and the carcinogenicity of *pks+ E. coli*.

## 1. Introduction

Colorectal cancer (CRC) is the third most common cancer worldwide [1]. Sporadic CRC (~85% of CRC) is associated with a Western lifestyle and microbiota dysbiosis [2,3]. CRC-promoting bacteria, such as *Bacteroides fragilis*, *Enterococcus faecalis,* and some strains of *Escherichia coli*, have been described as “drivers” capable of promoting the early stages of tumors development [4]. *E. coli* strains isolated from colon tissues of CRC patients frequently harbor the *pks* genomic island (*pks+ E. coli*) [5,6,7]. In various CRC mouse models, *pks+ E. coli* increased the number of tumors [8,9]. These *E. coli* induce double-strand DNA breaks and gene mutagenesis in intestinal epithelial cells [5,10]. A distinct mutational signature of *pks+ E. coli* was found following infection of human intestinal organoids, which was also detected in a subset of CRC patients [11]. *Pks+ E. coli* are therefore suspected of promoting colon cancer. The *pks* island encodes a complex biosynthesis pathway that enables the synthesis of the genotoxin, colibactin [12]. This metabolite is unstable, and the genotoxic effects of *pks+ E. coli* occur only upon direct contact between live bacteria and epithelial cells [5,10,13]. A recent study has revealed that adhesion-mediated epithelial association plays a crucial role in colibactin-mediated DNA damage and CRC exacerbation [14]. This bacterial adhesion to host epithelial cells is facilitated by the type 1 pilus adhesin FimH and the F9 pilus adhesin FmlH. Therefore, the effect of colibactin appears to be contact dependent and may be reduced by the presence of the protective mucus layer covering the digestive tract [15,16,17]. Recently, Harnack et al. have shown that, when the protective mucus layer of mice is disrupted by dextran sulfate sodium (DSS), the distance between *pks+ E. coli* and epithelial cells is shorter compared to infected mice without DSS [17]. In addition, the disruption of the mucus barrier was associated with an increase in DNA damage and extensive injury to the colon epithelium. Consequently, the integrity of the mucus layer appears to be paramount in order to safeguard against the detrimental effects of colibactin, and it may serve to impede the development of tumorigenesis in the colon. As a consequence of mucus depletion in Muc^2−/−^ mice, Velcich et al. observed the development of adenomas in the small intestine of these mice that progressed to invasive adenocarcinoma [18].

Pathogens have developed specific mechanisms to subvert and penetrate the mucus barrier. The serine protease autotransporters of enterobacteria (SPATEs) from class 2 have been identified in pathogenic *E. coli* to degrade several mucins through mucinase activity [19]. StcE is a metalloprotease described in enterohaemorrhagic *E. coli* O157:H7 that contributes to the attachment of these bacteria to the intestinal epithelium [20,21]. The EatA autotransporter found in enterotoxigenic *E. coli* is involved in mucin degradation, and this participates in the delivery of *E. coli* toxins to the cell surface [22]. Another example is the autotransporter Pic found in enteroaggregative *E. coli* and *Shigella flexneri*. Pic protease can degrade a variety of mucins (MUC2 and MU5AC) and enhances mice colonization by these strains [23]. The adherent and invasive *E. coli* strain LF82 produces a SPATE called Vat that is implicated in the degradation of mucins and therefore decreases mucus viscosity. The Vat protein was firstly described to cause cell cytoskeleton changes and vacuole formation in uropathogenic infection by *E. coli* [24]. Vat is 79% similar to Hbp/Tsh mucinases [25], which have also mucinase activity [26,27]. Nevertheless, the presence of *pic, vat,* and *tsh*/*hbp* genes was positively associated with the phylogroup B2 and extraintestinal pathogenic *E. coli*. We wondered whether these proteases were present in *pks+ E. coli*, whether they could promote the crossing of the intestinal mucus layer, and whether they could ultimately induce DNA damage.

## 2. Results

### 2.1. Vat Gene Predominates in pks+ E. coli

To identify virulence genes, including mucin proteases that are common to *pks+ E. coli*, the RefSeq and the VirulenceFinder databases were employed. Aside from adhesion factors, siderophores, and toxins, only three genes encoding mucin proteases were identified among the 2041 genomes of *pks+ E. coli* (Appendix A). The *vat* gene exhibited the highest prevalence (92%), followed by *pic* (42%) and *hbp*/*tsh* (4%) (Figure 1A and Appendix A). The current laboratory possesses a bank of *E. coli* strains (*n* = 222) isolated from tumoral colonic biopsies (*n* = 112) of CRC (*n* = 87) or control patients (*n* = 25), and this bank has been described in previous studies [6]. Following PCR screening of these strains for the three mucinases, it was found that the prevalence of *vat* was significantly higher in CRC patients (56%, *n* = 49/87) than those isolated from control (24%, *n* = 6/25, *p* = 0.0044) (Figure 1B). The prevalence of *pic* was not significantly different in CRC patients (22%, *n* = 19/87) compared to the control group (12%, *n* = 3/25), and *hbp*/*tsh* was not found in the clinical isolates. Then, we focused on the association between the *vat* gene and the *pks* genomic island in CRC patients (Figure 1C). Among patients with *pks-* strains (48%, *n* = 42/87), 18% of *E. coli* were found to be positive for the *vat* gene. Conversely, in patients with *pks+ strains* (52%, *n* = 45/87), the *vat* gene was present in all *pks+ E. coli.* This co-association suggests that the Vat mucinase may play a key function in the pathophysiology of *pks+ E. coli.*

### 2.2. Vat Protease Promotes Mucosal Crossing of pks+ E. coli

Vat promotes crossing of the intestinal mucus layer by adherent and invasive *E. coli* strains [25]. A column mucin gel penetration assay was performed with an *E. coli pks + vat+* (the POP198 clinical strain), its isogenic mutant *E. coli pks + vat-*, and the trans-complemented *E. coli pks + vat-* +*pBAD-vat* strain (Figure 2A). Unlike *E. coli pks + vat-*, the wild-type strain and the trans-complemented strain penetrated to the bottom of the mucus column (fraction n°1). In fraction n°2, *E. coli pks + vat-* was approximately two orders of magnitude less efficient at penetrating the mucus column than the *E. coli pks + vat+* and *E. coli pks + vat-* +*pBAD-vat* strains. To confirm that Vat enhances the penetration of *pks+ E. coli* in mucus, we infected human colonic epithelial cells, HT29-16E, which produce mucin of the colonic mucus layer as MUC2 [28]. The distance between bacteria and the epithelial cell surface was determined using the Cy3*-pks+ E. coli* probe. Vat-producing *E. coli* were found in closer proximity to the cells than the *vat*-deleted mutant (mean 2.6 ± 0.6 and 2.7 ± 0.3 µm for *E. coli pks + vat+* and *E. coli pks + vat- +pBAD-vat*, respectively, and 3.8 ± 0.5 µm for *E. coli pks + vat-*). The proportion of bacteria present in the immediate vicinity of the mucus (<2 µm) was 37 ± 13% for *E. coli pks + vat+*, 33 ± 4% for *E. coli pks + vat- +pBAD-vat,* and 16 ± 7% for *E. coli pks + vat-* (Figure 2B). A significant proportion (44%) of the bacteria deleted for *vat* were observed at a distance of greater than 4 µm. Overall, *E. coli pks + vat+* and *E. coli pks + vat-* +*pBAD-vat* strains were closer to the HT29-16E cells than the *E. coli pks + vat-* mutants, as illustrated in Figure 2C.

The CRC-associated *pks+ E. coli* strain 11G5 has been used in several *pks+ E. coli* studies [14,29,30]. To complete our results, we generated a mutant deleted for *vat* in this strain. The results obtained were in line with those described above (Appendix A). Thus, the Vat protease could promote the spread of *pks+ E. coli* in mucus.

### 2.3. Vat Protease Enhances the Genotoxic Effects of pks+ E. coli

As *pks+ E. coli* induce DNA damage in infected cells, we next speculated whether Vat, by improving *pks+ E. coli* access to epithelial cells, would increase the genotoxicity of bacteria [5,10]. HT29-16E cells were infected with *E. coli pks + vat+, E. coli pks + vat-*, or *E. coli pks + vat-* +*pBAD-vat,* and the fluorescence intensity of phosphorylated γ-H2AX foci was measured, as it is a well-known marker of DNA double-strand-breaks [31]. Cells infected with the *E. coli pks + vat+* strain displayed a significant increase in DNA damage, exhibiting approximately 2-fold greater fluorescence intensity of phosphorylated γ-H2AX foci compared to cells infected with *E. coli pks + vat-* (mean: 1.5 for *E. coli pks + vat+* compared to 0.7 for *E. coli pks + vat-*, *p* < 0.05). The signal intensity was 3-fold greater (mean: 2.2) with the *E. coli pks + vat-* +*pBAD-vat* strain (Figure 2D,E). A significant decrease in the fluorescence intensity signal of phosphorylated γ-H2AX foci in infected HT29-16E was also observed with the *pks+ E. coli* strain 11G5 with *vat* deleted (Appendix A). By facilitating *pks+ E. coli* crossing, Vat protease permits bacterial encroachment, thus enabling the close proximity of bacteria and epithelial cells and plays a role in the genotoxicity of *pks+ E. coli*.

### 2.4. Vat Protease Contributes to pks+ E. coli Colonization and Tumorigenesis in Apc^Min/+^ Mice

Regarding such capacity of *E. coli pks + vat+,* we decided to investigate whether Vat protease could contribute to the tumorigenesis capacity of *pks+ E. coli* in a CRC mouse model. The *Apc*^Min/+^ mouse model is commonly used to study multiple intestinal neoplasia in mice. A nonsense mutation in the *Apc* tumor suppressor gene leads to the spontaneous development of tumors [32]. Thus, *Apc*^Min/+^ mice were orally challenged with 10^9^ CFU of *E. coli pks + vat+*, *E. coli pks + vat-*, or a saline solution (PBS) and colonization was followed by quantitative bacteriological fecal analyses (Figure 3A). The colonization of *pks+ E. coli* was significantly reduced when *pks+ E. coli* was deleted for the *vat* gene. We observed a 1-log (day-14) to 3-log (day-42) decrease in colonization between mice infected with *E. coli pks + vat-* compared to those infected with *E. coli pks + vat+* (Figure 3B,C). At the completion of the infection period, a 5-log reduction in *E. coli pks + vat-* was isolated from mucosal colon tissue than *E. coli pks + vat+* (Figure 3D). This significant decrease in colonization of the intestine by *E. coli pks + vat-* was associated with a significant reduction in the number of colonic tumors (Figure 3E). These results were also found using the wild-type 11G5 strain and the associated *vat* mutant (Appendix A).

## 3. Discussion

The *vat* gene was found to be present in all *pks+ E. coli* strains isolated from CRC patients. As demonstrated here, Vat protease facilitates the crossing of *pks+ E. coli* in the mucus layer, enabling access to epithelial cells and resulting in DNA damage. This gain in accessibility to the epithelium has been reported in other bacteria. For example, StcE is a metalloprotease described in enterohemorrhagic *E. coli* O157:H7 that contributes to their attachment to the intestinal epithelium [20,21]. The EatA autotransporter found in enterotoxigenic *E. coli* is involved in mucin degradation, allowing the delivery of *E. coli* toxins to the cell surface [22]. We report that Vat is also a factor of colonization for *pks+ E. coli.* It has been previously reported that Vat contributes to gut colonization of adherent-invasive *E. coli*; however, the colonization has been evaluated on the 3rd day [25]. Our study demonstrated that Vat provides a persistent colonization advantage, even 44 days after infection. Pic protease degrades mucins and promotes intestinal colonization of enteroaggregative *E. coli* over a period of 14 days [23]. Our results complement these studies and suggest that Vat, and probably other mucin proteases, are implicated in the colonization of *E. coli* in the intestine. By facilitating colonization of *pks+ E. coli*, Vat could promote tumorigenesis of *pks+ E. coli* in the absence of other factors altering the mucus layer.

## 4. Materials and Methods

### 4.1. Clinical E. coli Isolates

Clinical *E. coli* strains were isolated from biological samples of patients from the University Hospital of Clermont-Ferrand, with ethical approval for human study no. DC-2017-2972. Biological samples were collected from CRC patients or controls patients (surgery for benign diverticulosis). All patients underwent surgery for resectable CRC at the Digestive and Hepatobiliary Surgery Department of the University Hospital of Clermont-Ferrand [33]. The exclusion criteria included clinically suspected hereditary CRC based on the revised Bethesda criteria, administration of neoadjuvant chemotherapy, a history of previous colonic resection, emergency surgery, and use of antibiotics within 4 weeks before the surgery. Biological samples were treated, and *E. coli* were collected as described previously [6].

### 4.2. Bacterial Strains and Construction of Isogenic Mutants

The clinical *E. coli* POP198 and 11G5 (*pks + vat+*) strains were isolated from hospital patients treated at Clermont-Ferrand (Appendix A). *Vat* isogenic mutants were generated using the method described by Datsenko et al. [34] and modified by Chaveroche et al. [35]. Briefly, the spectinomycin resistance cassette was used to replace the *vat* gene in *E. coli* POP198 or 11G5 cells with specific primers (Appendix A). To obtain trans-complemented strains, the *vat* gene was cloned into the pBAD33.1 plasmid (Appendix A) using the In-Fusion^®^ HD Cloning system (Takara, Saint-Germain-en-Laye, France) and electroporated in the POP198Δ*vat* and 11G5Δ*vat*. For experiments, strains were grown in Lysogeny Broth (LB) overnight at 37 °C with shaking at 110 rpm. Depletion of *vat* did not affect the bacterial growth (Appendix A). All strains used for this study are summarized in Appendix A.

### 4.3. Metagenomics Analysis and Screening of Mucin Proteases and pks Genes

The RefSeq database (v1.214 databases) and VirulenceFinder 2.0 tool were used to determine the most prevalent virulence genes among *pks+ E. coli*. RefSeq *E. coli* genomes were blasted against the clbQ gene, which is always present inside the *pks* island (clbQ is required to produce colibactin). Among the 2041 *pks+ E. coli* genomes selected, virulence genes were identified using the VirulenceFinder 2.0 tool and were classified into five proteins families based on their biological functions.

PCR was performed using the primers listed in Appendix A to screen for the presence of mucin proteases or the *pks* island in *E. coli* from hospital patients treated at Clermont-Ferrand (*n* = 222 strains) as described previously [36].

### 4.4. Mucin Column Penetration Assay

The mucin column penetration assay was used as previously described [25]. Briefly, a solution containing 10% mucin of porcine stomach (Merck, Darmstadt, Germany) and 0.3% agarose in HBSS was loaded into a 1.2-mL injection syringe, creating a mucous column. Briefly, 0.1 mL of prepared bacterial cells (10^9^ cells/mL) was layered onto the mucin. The columns were incubated for 3 h at 37 °C in a vertical position. Afterwards, four fractions (each one contains 0.3 mL) were collected from the button by applying gentle pressure. Each fraction was serially diluted and plated onto Columbia blood medium for CFU (colony-forming unit) enumeration.

### 4.5. Cell Culture

Human colon adenocarcinoma HT29-16E (MUC2 mucus producing) cells were cultured as previously described [28]. For infection experiments, cells were seeded at a density of 2 × 10^5^ cells per cm^2^ in 24-well culture plates with coverslips for 21 days to allow mucin production. Cells were infected with *E. coli* at a multiplicity of infection (MOI) of 100 for 45 or 90 min. Immunofluorescence and fluorescence in situ hybridization (FISH) experiments were performed at >21 days after HT29-16E culture and 24 h post-infection.

### 4.6. Fluorescence In Situ Hybridization (FISH)

To visualize *E. coli* at the end of 45 min of infection, HT29-16E infected cell wells were washed and fixed for 24 h with a fixative solution of Poloxamer 407 (16758, Merck, Darmstadt, Germany) at 20% (*w*/*w*) in 10% NBF (neutral buffered formalin, Sigma-Aldrich, HT501128) solution to enhance mucus structure integrity during fixation and to initiate polymerization at room temperature as published by Macedonia et al. [37]. Coverslips were washed in 100% methanol and conserved at −20 °C or directly used with the following procedures. Cells were washed with PBS with hybridization solution (20 mM Tris-Hcl, pH 7.4, 0.9 M NaCl, 0.1% (*w*/*v*) SDS) before incubation with 5 ng/µL of Cy3-*E. coli* probes (Appendix A) [38]. Paraffin was used to prevent drying, and coverslips were incubated at 46 °C for 3 h in a humidified chamber. Slides were washed in PBS 3X for 5 min before nuclei staining with Hoechst 33342 (Sigma-Aldrich, B2261) and mucus staining with 647-wheat germ agglutinin (ThermoFisher Scientific, W32466, Illkirch-Graffenstaden, France). Coverslips were mounted using Mowiol preparation (494338, Dutscher, Bernolsheim, France). Images were taken using a Zeiss LSM 980 confocal microscope (Zeiss, Voisins-le-Bretonneux, France). To evaluate the distance between bacteria and epithelial cells, Imaris software (v10.0.1). was used, and the bacteria count was determined for each 2 µm mucus layer in each field.

### 4.7. Immunofluorescence

For γ-H2AX immunofluorescence staining, following infection, cells were washed using PBS, then post-infection media, namely, Dulbecco’s Modified Eagle Medium (Gibco, Waltham, MA, USA) culture media supplemented with gentamicin (200 µg/mL), was added. After 24 h in post-infection media, cells were fixed with PBS with 4% formaldehyde for 10 min, permeabilized in PBS with 0.25% Triton X-100 for 10 min, and blocked with PBS with 5% fetal calf serum for 30 min. Anti-γH2AX primary antibodies (9718, Cell Signaling, Danvers, MA, USA) were diluted 1/400 and incubated overnight at 4 °C. The secondary antibody used was conjugated to Alexa 488 (ThermoFisher, A-21206). Nuclei were stained with Hoechst 33342 (Sigma-Aldrich, B2261), and mucus was stained with 647-wheat germ agglutinin (ThermoFisher, W32466). Coverslips were mounted using Mowiol preparation (Dutscher, 494338). Images were taken using a Zeiss LSM 980 confocal microscope as HT29-16E cells form a multilayer after 21 days of culture. To evaluate the percentage of γH2AX-expressing cells, the fluorescence intensity of γH2AX foci was determined with Imaris software and expressed as the Hoechst fluorescence intensity for each field counted.

### 4.8. Apc^Min/+^ Mouse Model Information and Quantification of E. coli in Stools and Colonic Tissues

C57BL/6 *Apc*^Min/+^ females 6 weeks of age were inoculated with oral administration of 10^9^ CFU of *E. coli pks + vat+* or *E. coli pks + vat-* mutants, as previously described [30]. For the control group, a saline solution (PBS) was administered. Same litter mates were housed together in individually ventilated cages with seven to eight mice per cage. All mice were maintained on a regular diurnal lighting cycle (12:12 light:dark) with ad libitum access to food and water. For *E. coli* quantification, feces were collected periodically, crushed (Ultra-Turrax, IKA, Staufen, Germany) in PBS, and spread on chromID CPS Elite agar (BioMérieux, Craponne, France) plates, which allowed the detection of *E. coli*. A random selection of 10 *E. coli*/mouse was analyzed with PCR using specific primers located in the *clbQ* gene of the *pks* island and in the *vat* gene. No detection of *E. coli pks + vat+* was observed in PBS mouse control groups. *E. coli* isogenic mutant was detected with spectinomycin resistance (this resistance gene was inserted into *E. coli* genome when *vat* was deleted) in LB plates. Colonic tissues were collected, flushed with PBS to remove most of the fecal bacteria, and then treated as feces. All mice were sacrificed 44 days (POP198) or 56 days (11G5) post-infection. Colonic tumor numbers were determined using a dissecting microscope.

Animal protocols were performed in accordance with French and European Economic Community guidelines (86-60, EEC) for the care of laboratory animals. The study was approved by the French Ministry of Higher Education Research and Innovation (Apafis no. 22798).

### 4.9. Statistical Analysis

Unpaired Student *t*-tests and Mann–Whitney tests were used for comparisons of two groups where appropriate (normality verified by the Shapiro–Wilk test and homoscedasticity by the FisherSnedecor test). For multiple groups, one-way non-parametric analysis or post-hoc multiple comparison tests were used and analyzed with GraphPad Prism version 6.07 software. *p* < 0.05 was considered significant.

## 5. Conclusions

In summary, we first described the Vat mucin protease in the *pks+ E. coli* pathogenic bacteria. Our discovery contributes to the understanding of the pathophysiology of *pks+ E. coli* strains. Mucus crossing is the step prior to adhesion to epithelial cells, which is critical for colibactin to exert its genotoxic activity.

## Figures and Tables

**Figure 1 ijms-26-05353-f001:**
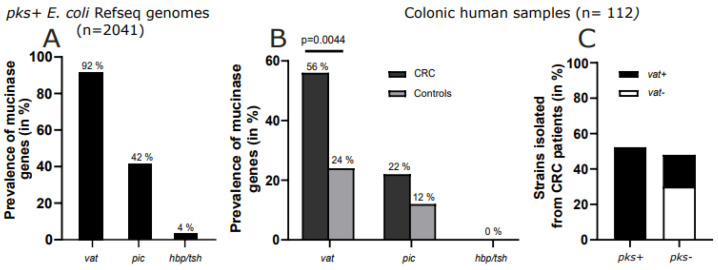
Mucin protease prevalence in *pks+ E. coli*. (**A**) Metagenomics analysis of most frequently found mucin proteases in *pks+ E. coli* (*n* = 2041). *E. coli* sequences in databases were obtained from different metagenomic analysis, and *vat* prevalence was searched among *pks+ E. coli* (*clbQ* screening). (**B**) Mucin protease prevalence in *E. coli* isolated from CRC patients (*n* = 112). Genes encoding *Vat, pic,* and *hbp*/*tsh* mucin proteases were screened using PCR in *E. coli* (*n* = 222) isolated from tumoral colonic human biopsies (*n* = 112) of CRC (*n* = 87) or normal mucosa for control patients (*n* = 25). Statistical analysis was performed with the χ^2^ test. (**C**) The *Vat* gene was screened with PCR among *pks+ E. coli* (*n* = 58) or *pks- E. coli* (*n* = 54) strains isolated from CRC colonic human biopsies. A patient was considered positive if at least one *pks+ E. coli* was detected in his or her biopsy.

**Figure 2 ijms-26-05353-f002:**
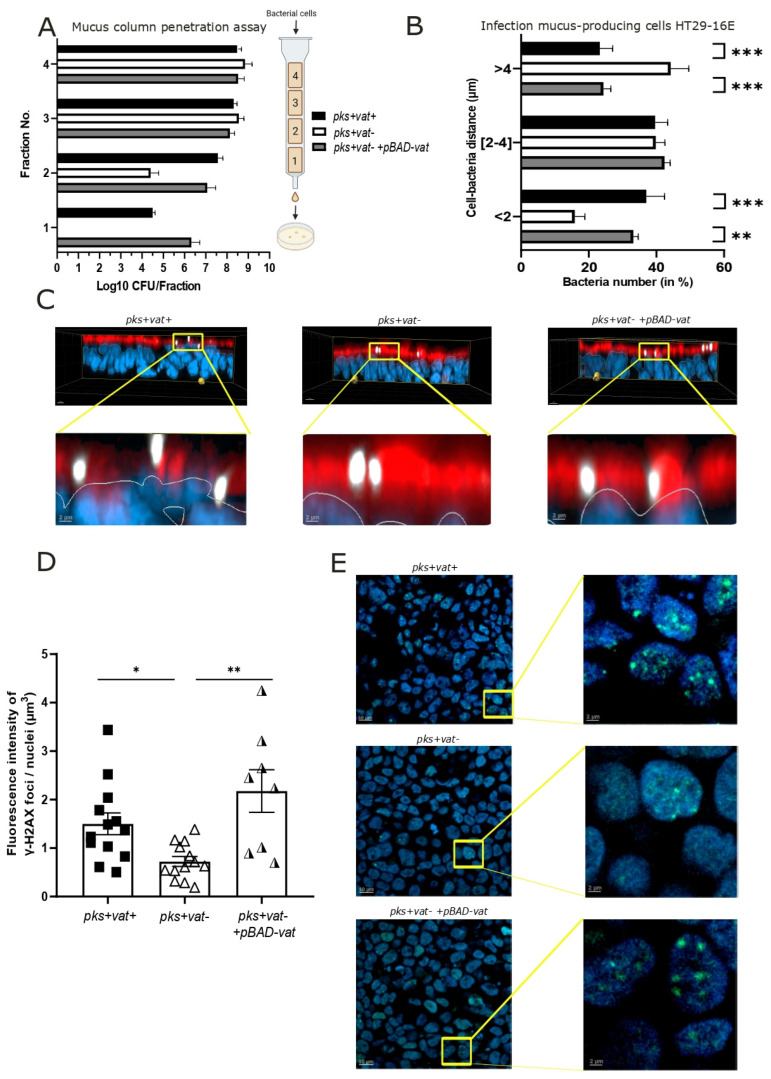
Deletion of the *vat* gene in *pks+ E. coli* impairs bacteria mucus crossing and reduces genotoxicity. (**A**) The assessment of mucus crossing using the column penetration assay. Quantification of indicated *pks+ E. coli* strains in fractions eluted from columns filled with gel-forming mucus (fractions 1 to 4 were sequentially obtained from the bottom to top of the column). Data are representative of three independent experiments. (**B**,**C**) Mucus-producing HT29-16E cells were infected for 45 min with a multiplicity of Infection of 100. For each replicate, three representative fields were analyzed, and experiments were performed three times. (**C**) Cells were stained with Hoechst (blue), mucus with WGA-leptin (red), and *pks+ E. coli* with fluorescence in situ hybridization using the Cy3*-pks+ E. coli* probe. Representative confocal microscopy images of infected HT29-16E cells are shown. (**B**) Cy3-*pks+ E. coli* (shown in white in (**C**)) were counted and the distance to Hoechst-stained cells was measured using Imaris software (v10.0.1). The total number of bacteria detected was *pks + vat*- *+pBAD-vat* (*n* = 1057), *pks + vat+* (*n* = 586), and *pks + vat-* (*n* = 377). The distances between bacteria to cell were determined for each field. Bacteria were counted in each 2 µm layer and then expressed as a percentage relative to the total number of bacteria for each condition. (**D**,**E**) Mucus-producing HT29-16E cells were infected for 90 min with a multiplicity of infection of 100. For each replicate, at least three representative fields were analyzed, and experiments were performed three times. Phosphorylated γ-H2AX foci fluorescence intensity was determined based on immunofluorescence of the nuclei volume (Hoechst staining) 24 h post-infection. Data are presented as means ± SEMs. Statistical comparisons were carried out by one-way Kruskal–Wallis nonparametric tests followed by Dunn posttests after normality testing (* *p* < 0.05, ** *p* < 0.01, *** *p* < 0.001).

**Figure 3 ijms-26-05353-f003:**
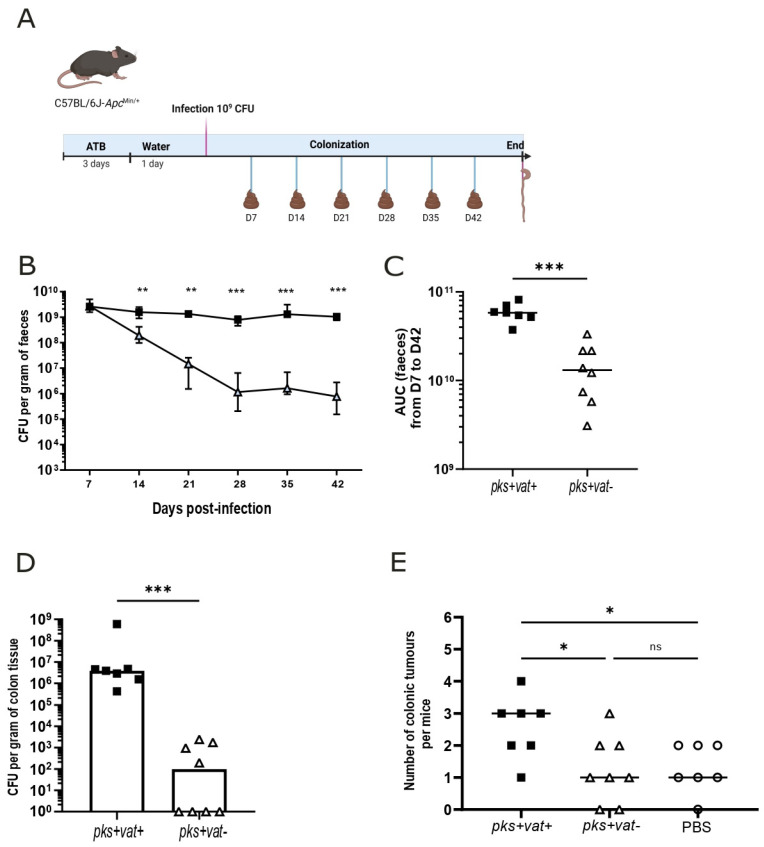
Vat favors gut colonization of *pks+ E. coli* and increases colonic tumors numbers in *Apc^Min^*^/*+*^. (**A**) *Apc^Min^*^/*+*^ mice were orally administered 10^9^ colony-forming units (CFUs) of *E. coli pks + vat+, E. coli pks + vat-* mutant, or PBS. In order to facilitate implantation of *E. coli*, streptomycin (2.5 g/L) was administered for 3 days prior to oral inoculation with bacteria. Feces were collected each once a week, and *E. coli pks + vat+* or *E. coli pks + vat-* CFUs per gram of feces were determined. (**B**) Bacterial colonization in the stools of mice from 7 to 42 days post-infection. Values are presented as medians ± errors. (**C**) Area under the curve (AUC) values from bacterial colonization in the stools of mice from 7 to 42 days post-infection. The data points represent actual values for each individual mouse, and the bars indicate median values. (**D**) The number of *E. coli* associated with non-tumoral colonic tissue at 44 days post-infection was determined. Colonization data are presented as medians. (**E**) The number of colorectal tumors per mouse was determined using a dissecting microscope. Data are presented as medians. Statistical comparisons were carried out using one-way Kruskal–Wallis nonparametric test for three groups mice or the Mann–Whitney test analysis for two groups mice followed by the Dunn posttest after normality testing (* *p* < 0.05, ** *p* < 0.01, *** *p* < 0.001, ns = not significant).

## Data Availability

Data are available at https://doi.org/10.17632/k5vnjk7mrw.1.

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
