# Peer review of "Vat-Mediated Mucus Penetration Enables Genotoxic Activity of *pks+ Escherichia coli"

_ijms, 2025, doi:10.3390/ijms26115353_

Round 1
Reviewer 1 Report
Comments and Suggestions for Authors
see attached review document

Author Response
Thank you very much for taking the time to review this manuscript. Please find the detailed responses below and the corresponding revisions/corrections highlighted/in track changes in the re-submitted files.
Please see the attachment.

Reviewer 2 Report
Comments and Suggestions for Authors
I have read the manuscript titled “Vat-Mediated Mucus Penetration Enables Genotoxic Activity of pks+ Escherichia coli” by Chat et al. I have the research focus is relevant and findings might certainly constitute an advance in the field. However, I think some improvements are required before the article being suitable for publication.
Major comments:
- Absence of Vat in coli pks+ strains used in this work (in cultures or supernatants) should be demonstrated. In addition, presence of Vat should be demonstrated in the complemented strain.
- Results, figure 1C: I expected to see 100% for vat in pks+ It is necessary to show the vat-negative portion if prevalence (%) is shown?
Minor comments:
- Abstract, line 12: “abnormally colonized” does not necessarily indicates a higher proportion or quantity. Please consider rephrasing.
- Abstract, line 14: a mention stating that the mucus layer covers the apical surface of the epithelial cells might help non-experts readers for better comprehension.
- Introduction, line 33: “pks” should it be in italics?
- Introduction, lines 46-48: This sentence describes a normal state, not only the infected mice. Please consider rewriting.
- Introduction, line 60: “0157” It seems to be zero, it must be an “O” letter.
- Introduction, line 62: “involved” instead “involve”
- Introduction, line 72: “intrinsic” is not part of the acronym.
- Introduction, lines 72-74: It sounds like proteases might cause DNA damage themselves. Please rewrite.
- Results, figure 2E: Pictures should be bigger. Alternatively, additional panels with “zoom-in” portions could be included to see positive foci.
- Results, figure 3E: Tumors were detected in PBS-treated mice. Was that expected? A mention should be included in results or discussion.
- Discussion, line 193: I think “spread” is not the correct term. “Crossing” or equivalent might be used.
- Discussion, line 202: It is clear that the experiment lasted 44 days but the sentence suggests that evidence of the end of the colonization was obtained. Please consider rephrasing.
I suggested revision in several cases in "minor comments", previous sections.
Author Response

(The authors gave the same response as above.)

Round 2
Reviewer 2 Report
Comments and Suggestions for Authors
My comments were properly addressed by the authors. I have no additional comments.